Estimation methods for the ratio of medians of three-parameter lognormal distributions containing zero values and their application to wind speed data from northern Thailand

Maneerat Patcharee 1
Nakjai Pisit 2
Niwitpong Sa-Aat sa-aat.n@sci.kmutnb.ac.th 3
1 Department of Applied Mathematics, Uttaradit Rajabhat University , Uttaradit , Thailand
2 Department of Computer Sciences, Uttaradit Rajabhat University , Uttaradit , Thailand
3 Department of Applied Statistics, King Mongkut’s University of Technology North Bangkok , Bangkok , Thailand
Mahmood Haider
Electronic publication date: 2022 Oct 11
Publication date: 2022
Volume: 10
Electronic Location ID: e14194
Received 2022 Jun 16; Accepted 2022 Sep 15
Copyright: ©2022 Maneerat et al.
Copyright year: 2022
Copyright holder: Maneerat et al.
License: This is an open access article distributed under the terms of the Creative Commons Attribution License, which permits unrestricted use, distribution, reproduction and adaptation in any medium and for any purpose provided that it is properly attributed. For attribution, the original author(s), title, publication source (PeerJ) and either DOI or URL of the article must be cited.
License URL: https://creativecommons.org/licenses/by/4.0/

Keywords: Wind speed, Particulate matter, Confidence intervals, Ratio of medians, Fiducial, Normal approximation, Bayesian method

Funding: Office of the Permanent Secretary, Ministry of Higher Education, Science, Research and Innovation (OPS MHESI) RGNS 64–196 Thailand Science Research and Innovation (TSRI) and Uttaradit Rajabhat University This work (Grant No. RGNS 64–196) was supported by the Office of the Permanent Secretary, Ministry of Higher Education, Science, Research and Innovation (OPS MHESI), Thailand Science Research and Innovation (TSRI) and Uttaradit Rajabhat University. The funders had no role in study design, data collection and analysis, decision to publish, or preparation of the manuscript.

==============================
Wind speed has an important impact on the formation and dispersion of fine particulate matter (PM), which can cause several health problems. During the transition from the winter to the summer season in northern Thailand, the wind speed has been low for longer than usual, which has resulted in fine PM accumulating in the air. Motivated by this, we have identified a need to investigate wind speed due to its effect on PM formation and dispersion and to raise awareness among the general public. The hourly windspeed can be approximated by using confidence intervals for the ratio of the medians of three-parameter lognormal distributions containing zero values. Thus, we constructed them by using fiducial, normal approximation, and Bayesian methods. By way of comparison, the performance measures for all ofthe proposed methods (the coverage percentage, lower and upper error probabilities (LEP and UEP,respectively), and expected length) were assessed via Monte Carlo simulation. The results of Monte Carlo simulation studies show that the Bayesian method provided coverage percentages close to the nominal confidence level and shorter intervals than the other methods. Importantly, it maintained a good balance between LEP and UEP even for large variation and percentage of zero-valued observations. To illustrate the efficacy of our proposed methods, we applied them to hourly wind speed data from northern Thailand.

Introduction

Oxygen in the air is necessary for the survival of humans and other animals. How fast the air moves past a certain point is known as wind speed (measured in km/h), which is an important phenomenon in meteorology for monitoring and predicting weather patterns in a given area. It is reported daily along with the temperature, precipitation, and humidity for all provinces on the Thai Meteorological Department website. Importantly, low wind speeds during the transition from the winter to the summer season lead to increased fine particulate matter (PM) levels. In 2020, high PM2.5 (PM ≤ 2.5 µm) increased the incidences of several ailments (respiratory illness, allergic reactions in the eyes and nasal passages, etc.) in almost all of the regions in Thailand, especially in the northern region (Tanraksa & Kendall, 2020). Spikes in PM2.5 levels in the northern and northeastern regions of Thailand occur during the transition from the winter to the summer season (January–March) (Wipatayotin & Tangprasert, 2021). These reasons have led to our interest in estimating wind speeds to provide essential information on current PM2.5 levels based on historic data. The hourly wind speed data for Phitsanulok and Phayao provinces located in the northern region follow the assumptions for a three-parameter lognormal (TPLN) distribution containing zero values indicating no wind. By way of comparison, estimating the ratio of the median wind speeds in two areas is used as a starting point in this study. The median, a measure of central tendency, is the central value of a dataset (the midpoint of a distribution). Moreover, it is more efficacious to use the median than the mean in analyses when the distribution of data is skewed. In addition, the ratio of the medians of two datasets can be used to measure the difference between them.

A lognormal distribution is used to represent right-skewed data when the threshold parameter (the lower bound of the data) is equal to zero. The TPLN distribution first introduced by Aitchison & Brown (1963) is suitable for highly right-skewed data that do not fit a lognormal distribution because the threshold parameter is greater than zero. It has been used in hydrology (Burges, Lettenmaier & Bates, 1975; Charbeneau, 1978) and for the analysis of flood frequency (Singh & Rajagopal, 1986; Singh & Singh, 1987). In this study, zero values are included among both simulated and wind speed data that follow a TPLN distribution (i.e., a TPLN distribution containing zero values).

One of statistical inference methods is the parameter estimations, including the point and interval estimations with the best-known example of the latter being a confidence interval (CI). According to Casella & Berger (2002), the CI is a range of numbers containing the parameter of interest with the desirable level of confidence which is better than a point estimator. For this reason, the CI is focused in this study. The point parameter estimations of the TPLN distribution have been formulated and discussed by various authors. Cohen & Whitten (1980) modified the local maximum likelihood and the moment estimators for the mean, variance and threshold parameters by using the first, second and third order statistics. Next, the moment estimation has been developed by replacing the third moment in a function of the first order statistics, as previously described in Cohen, Whitten & Ding (1985). Later, Singh, Cruise & Ma (1990) compared the five methods, including the regular method of moments, the modified method of moment (Cohen & Whitten, 1980), the regular maximum likelihood estimate (MLE), the modified MLE, and entropy to estimate the parameters and the quantiles of the TPLN via Monte Carlo simulation.

In particular, some researchers have formulated methods for estimating the CIs for the parameters of the TPLN. Royston (1992) used the zero skewness method to estimate the threshold parameter and its certain functions, motivated by Griffiths (1980). After that, Pang et al. (2005) presented the Bayesian estimation using Markov chain Monte Carlo to approximate the coefficient of variation of three-parameter Weibull, lognormal and gamma distributions. Later, Basak, Basak & Balakrishnan (2009) made use of numerical methods (Newton–Raphson and EM algorithms) based on progressively Type-II censored samples from the TPLN for assessing the local and modified MLEs for the mean, variance and threshold parameters as well as the CIs for the threshold by using Monte Carlo simulations. Chen & Miao (2012) conducted the order statistics to construct the exact CIs and the exact upper CIs for the threshold parameter. Recently, Maneerat, Niwitpong & Nakjai (2022b) formulated the CIs for the median of the TPLN distribution based on bootstrapping, normal approximation (NA) and the generalized pivotal quantity. Maneerat, Nakjai & Niwitpong (2022a) also proposed Bayesian confidence intervals based on different noninformative priors for the delta-lognormal mean.

However, the CIs for the parameters of a TPLN distribution, especially when zero values are included along with the nonzero values, have not yet been formulated. Therefore, we herein propose fiducial, NA, and Bayesian methods for constructing CIs for the ratio of the medians of TPLN distributions containing zero values. We conducted a Monte Carlo study to assess their performances in terms of their coverage percentages (CPs), lower and upper error probabilities (LEP and UEP, respectively), and expected lengths. These methods were applied to estimate hourly wind speeds during the transition from the winter to the summer season in northern Thailand. This information could be used to help design green corridors and implement other policies to reduce PM levels in northern Thailand in the future.

Three-parameter lognormal distributions containing zero values and the parameter of interest

Let Xi = (Xi1, Xi2, …, Xini), i = 1, 2 be random samples drawn from a TPLN distribution containing zero values, denoted as TPLNZγi,μXi,σXi2,ρi where γi is the threshold parameter, μXi is the scale parameter (the mean of Xi), σXi2 is the shape parameter (the variance of Xi) and ρi is the proportion of nonzero values. For Xi > 0, random variable Yi = ln(Xi − γi) is normally distributed as NμXi,σXi2 if Xi is a lognormal random variable (Cohen & Whitten, 1980). Thus, the probability density function (pdf) of Xi is given by (1) hXi;γi,μXi,σXi2,ρi=1−ρi+ρiXi−γi−12πσXi21/2 exp−12σXi2lnXi−γi−μXi2,

where γi < Xi < ∞, which are bounded as follows: γi ≥ 0, 0 < μXi < ∞, σXi2>0 and 0 < ρi < 1. Otherwise, for Xi = 0, the pdf of W becomes hXi;γi,μXi,σXi2,ρi=1−ρi. The likelihood and log-likelihood functions of ω=γi,μXi,σXi2,ρi are respectively given by

(2) Lω|X=1−ρini0ρini12πσXi2ni1/2 ∏j=1ni1 exp−lnXij−γi−12σXi2lnXij−γi−μXi2

(3) lω|X=ni0 ln1−ρi+ni1 lnρi−ni12ln2πσXi2−∑j:xij>0 lnXij−γi−12σXi2 ∑j:xij>0lnXij−γi−μXi2

where ni1 is the number of nonzero values that are binomially distributed with sample size ni = ni0 + ni1; ρi is the proportion of nonzero values ; ni0 is the number of zero values. Eq. (3) is computed from the first derivative about γi,μXi,σXi2,ρi. The respective MLEs of γi,μXi,σXi2,ρi are solved by setting their first derivative to zero as follows:

(4) ∂lω|X∂γi= ∑j:xij>01Xij−γi+ni1 ∑j:xij>0lnXij−γi−1ni1 ∑j:xij>0 lnXij−γi2 ∑j:xij>01Xij−γilnXij−γi−1ni1 ∑j:xij>0 lnXij−γi=0

(5) ∂lω|X∂μXi=σXi−2 ∑j:xij>0 lnXij−γi=0

(6) ∂lω|X∂σXi2=−2σXi2−1ni1+2σXi4−1 ∑j:xij>0lnXij−γi−μXi2=0

(7) ∂lω|X∂ρi=ni01−ρ−1−ni1ρ−1=0.

It is difficult to determine the explicit form of γ ˆi from the Eq. (4), so the MLE of γi can be obtained by maximizing Eq. (4) using the Adam algorithm. It is a replacement optimization algorithm for stochastic gradient descent based on adaptive moment estimation that is mainly used in neural networks and other machine learning algorithms (Ruder, 2016). It provides a way of computing adaptive learning rates for specific parameters. Let β1 and β2 be the initial time step and decay rate, respectively. Adam performs well in practice when β1 = 0.9 and β2 = 0.999 (Singh, 2020). By optimizing γi, the decaying averages and the past squared gradients are the estimates of the first and second moments of the gradients as follows: (8) mt,vt=β1mt−1+1−β1gt,β2vt−1+1−β2gt2

where gt denotes the first derivative of the target function (4) at time t = 0. Note that (mt, vt) are initialized as vectors of 0’s, and the Adam algorithm operates until their gradients approach zero. The bias-corrected first- and second-moment estimates are used to update parameter γi as follows: (9) m ˆt,v ˆt=mt1−β1t,vt1−β2t.

Finally, the parameter γi is updated by Adam update rule, which is defined as (10) γi,t+1=γi,t−δv ˆt+ϵ−1m ˆt

where δ and ϵ denote the step size and the learning rate, respectively. ϵ is fixed at 10−8 for a sufficient learning rate using the Adam optimization algorithm, as can be seen in Algorithm 1.

After obtaining γ ˆi by using Adam, one can compute the MLEs of μXi,σXi2,ρi as follows: (11) μ ˆXi,σ ˆXi,mle2,ρ ˆi=ni1−1 ∑j:xij>0 lnXij−γ ˆi,ni1−1 ∑j:xij>0lnXij−γ ˆi−μ ˆXi2,ni1ni.

Since ηi = ρi[γi + exp(μXi)] provides the medians of Xi, the ratio of the medians of Xi (the parameter of interest in the this study) is given by (12) ω=η1/η2.

By substituting MLEs γ ˆi,μ ˆXi,σ ˆXi,mle2,ρ ˆi, estimate η ˆi=ρ ˆiγ ˆi+ expμ ˆXi becomes the MLE of ηi. The concepts are elaborated in the methods for constructing CIs for ω in the next section.

Methods

Here, we present constructing the CIs for the ratio of the medians in the TPLN models containing zero observations based on the fiducial methods (the fiducial generalized pivotal quantity (fiducial GPQ)) and the method of variance estimates recovery-based fiducial generalized pivotal quantity (MOVER-fiducial GPQ), NA (McKean & Schrader, 1984) and the Bayesian methods.

The fiducial method

Hannig, Iyer & Patterson (2006) described the fiducial method based on the fiducial GPQ, a GPQ subclass introduced by Weerahandi (1993). Fiducial techniques have been used in constructing CIs in several research studies (Kharrati-Kopaei, Malekzadeh & Sadooghi-Alvandi, 2013; Li, Zhou & Tian, 2013; Maneerat, Niwitpong & Niwitpong, 2020). Let Y = (Y1, Y2, …, Yn) be a random variable with probability density function fY(y; λ, θ), where λ and θ are vectors of the parameter of interest and the nuisance parameter, respectively. Moreover, let y = (y1, y2, …, yn) be the observed values of Y, and assume that fiducial GPQ T(Y; y, λ, θ) is only a function of λ. This method is especially associated with the fiducial inference proposed by Fisher (1935). The fiducial GPQ CI depends on the fiducial GPQ defined by Hannig, Iyer & Patterson (2006) in Definition 1. Recently, Chankham, Niwitpong & Niwitpong (2022) recommended the fiducial GPQ-based CI for estimating the coefficient of variation of an inverse gaussian distribution when the sample size was small. Similarly, the performance of a fiducial GPQ CI in terms of expected length was the shortest when used to estimate the common coefficient of variation of delta-lognormal distributions by Yosboonruang, Niwitpong & Niwitpong (2022).

Definition 1.

A GPQ T(Y; y, λ, θ) for a parameter λ is called a fiducial generalized pivotal quantity (FGPQ) if it satisfies the following conditions: (FGPQ1) Given Y = y, the T(Y; y, λ, θ) distribution is free of all parameters. (FGPQ2) For every y ∈ R+, the observed pivotal T(Y; y, λ, θ) = λ.

If the T(Y; y, λ, θ) satisfies the conditions (FGPQ1) and (FGPQ2), then it is possible to construct the 100(1 − φ)% fiducial GPQ-based CI for λ is [Tλ(φ/2), Tλ(1 − φ/2)]; Tλ(φ) is the φth percentile of T(Y; y, λ, θ).

The fiducial GPQ CI

The fiducial GPQ CI can be constructed based on the fiducial GPQ conditions in Definition 1. Recall that (γi,μXi,σXi2,ρi) are the parameters controlling the behavior of a TPLN containing zero observations. Motivated by the possible value γi < Xi(1), we proposed the fiducial GPQ of γi based on a continuously uniform distribution and the fiducial GPQ of ρi based on a beta distribution, respectively, as follows:

(13) Tγi∼Uniformmin=0,max=Xi1

(14) T1−ρi∼Betani0+1,ni1+1

where Xi(1) denotes the minimum value of Xi and ni1 is the sample size of nonzero values ni = ni1 + ni0. Furthermore, the fiducial GPQ of μXi can be obtained by using the concepts of Krishnamoorthy & Mathew (2003) as (15) TμXi=μ ˆXi−WiTσXi2/ni1

where TσXi2=ni1−1σ ˆXi2/Vi is the fiducial GPQ of σXi2. Moreover, estimator σ ˆXi2=ni1−1−1∑j:xij>0 lnXij−γ ˆi. Note that the random variables Wi and Vi are independently draw from a standard normal distribution and a chi-square distribution with ni1 − 1 degrees of freedom, respectively. Since the three fiducial pivots (Tγi, Tρi, TμXi) satisfy the fiducial GPQ properties and Tηi = T1−ρi[Tγi + exp(TμXi)] is the fiducial GPQ of ηi, then we can obtain the the 100(1 − φ)% fiducial GPQ CI for ω as follows: (16) lωF,uωF=Tωφ/2,Tω1−φ/2

where Tω = Tη1/Tη2, and Tω(φ) denotes the φth percentile of Tω. The steps for calculating the CP of fiducial GPQ CI (CPF) for ω can be carried out by using Algorithm 2.

The MOVER-fiducial GPQ CI

MOVER is a well-known method for estimating the CI of the parameter of interest (Donner & Zou, 2012; Harvey & Van der Merwe, 2012; Hasan & Krishnamoorthy, 2017; Maneerat & Niwitpong, 2020; Maneerat, Niwitpong & Niwitpong, 2021; Zhang et al., 2021; Maneerat, Nakjai & Niwitpong, 2022a). Moreover, it can produce an explicit form of the CI that is easy to compute. For these reasons, we derived the MOVER-fiducial GPQ CI for ω as follows:

The fiducial GPQ CIs for (γi, μXi, ρi) can respectively be written as

(17) lγi,uγi=Tγiφ/2,Tγiφ/2

(18) lμXi,uμXi=TμXiφ/2,TμXiφ/2

(19) lρi,uρi=Tρiφ/2,Tρiφ/2.

These intervals can be formulated the MOVER CI by using the concept of Donner & Zou (2012) such that the 100(1 − φ)% MOVER-fiducial CI for lnηi becomes (20) llnηi,ulnηi=lnη ˆi−lnρ ˆi−lγi2+lnγ ˆi+ expμ ˆXi−llnγi+expμXi2,lnη ˆi+lnρ ˆi−uγi2+lnγ ˆi+ expμ ˆXi−ulnγi+expμXi2

where

(21) llnγi+expμXi= lnγ ˆi+ expμ ˆXi−γ ˆi−lγi2+expμ ˆXi−explγi2

(22) ulnγi+expμXi= lnγ ˆi+ expμ ˆXi+γ ˆi−lγi2+expμ ˆXi−explγi2,

thereby providing the 100(1 − φ)% CI-based the MOVER-fiducial GPQ CI for ω as (23) lωM,uωM= explnη ˆ1− lnη ˆ2−lnη ˆ1−llnη ˆ12+lnη ˆ2−llnη ˆ22,lnη ˆ1− lnη ˆ2+lnη ˆ1−llnη ˆ12+lnη ˆ2−llnη ˆ22.

Algorithm 3 presents the computational steps for calculating the CP of the MOVER-fiducial GPQ CI (CPM) for ω.

The NA method

According to probability theory, the concept behind this method is the assumption that the approximate distributions of all of the samples approach a normal distribution pattern if the sample size is sufficiently large. This idea is integrated with the central limit theorem, in which the distribution of a given sample mean is approximated as a normal pattern if the sample size is sufficiently large under the assumption that all of the samples are similar to each other regardless of the shape of the population distribution. Recently, Maneerat, Niwitpong & Nakjai (2022b) proposed an NA-based CI for the median of a TPLN distribution, which performed well for a large sample size. Thus, we also considered the NA method.

Given a set of observations, threshold γi can be estimated by using the Adam algorithm to find the MLEs of the mean and variance (μ ˆXi,σ ˆXi2). Here, the medians of TPLN models with zero observations can be log-transformed to become (24) lnηi= lnρi+ lnγi+ expμXi

which can be approximated by using (γ ˆi,μ ˆXi,ρ ˆi) to give lnη ˆi= lnρ ˆi+ lnγ ˆi+ expμ ˆXi. Using the delta method, the variance of lnρi can be derived as (25) Vlnρ ˆi=1−ρinρi.

Likewise, McKean & Schrader (1984) estimated the variance of the median as a distribution-free estimate defined as follows: (26) Vη ˆiMS=Xini1−c+1+Xic21.962

where Xi(r) denotes the rth order statistic in a random sample drawn from a TPLN model with zero observations of size ni1, and c=ni1+1/2−1.96ni1/4. Later, Hettmansperger & Sheather (1986) claimed that VMS is a consistent estimator of the variance of the median. Similarly, the variance of lnη ˆi can be derived by applying the delta method as follows: (27) Vlnη ˆi≅Vη ˆiMSEη ˆi2

where Eη ˆi=γ ˆi+ expμ ˆXi+σ ˆXi22ni1−1 is the expectation of η ˆi. Estimated variance V ^lnη ˆi is obtainable by replacing γ ˆi,μ ˆXi,σ ˆXi2,ρ ˆi from the sample. Thus, by applying the central limit theorem , the random variable Wi can be defined as (28) Wi=lnη ˆi− lnηiV ^lnη ˆi

which approaches a standard normal distribution as n → ∞. Subsequently, the 100(1 − φ)% NA-based CI for ω can be written as (29) lωN,uωN= explnη ˆ1− lnη ˆ2∓W1−φ/2V ^lnη ˆ1+V ^lnη ˆ2

where Wφ denotes the φth percentile of a standard normal distribution. Algorithm 4 was used to compute the CP of the NA (CPN).

The Bayesian method

Bayesian methods are based on treating probability as beliefs rather than frequencies. Given unknown parameter θ, a prior distribution p(θ) represents the subjective belief as a subjective distribution formulated before the data are seen. The posterior distribution is obtained from a prior that is updated with the likelihood function (or sample information) by using Bayes’ rule (Casella & Berger, 2002). Importantly, the posterior distribution is considered to be a random quantity and can be used to make a statement about θ, For example, the point and interval estimates of θ can be computed by using its posterior. Equal-tailed Bayesian intervals based on the Jeffreys’ and uniform priors based on the posterior densities of the zero proportion and the variance have been shown to perform well in certain scenarios (Yosboonruang, Niwitpong & Niwitpong , 2022).

Here, the Bayesian CI for ω is formulated based on the Bayesian method. First, we define an informative prior for our objective assumption depending on the amount of information available in the data as (30) Pγ1i,μXi,logσXi,ρi=constant∗1−ρiαi−1ρiβi−1Γαi+βiΓαiΓβi−1.

The posterior densities of (γi, ρi) are obtained by obtaining Eq. (30) with the likelihood function (2) as

(31) fγi=Pγi|X,μ ˆXi,σ ˆXi2=constant∗1−ρiαi−1ρiβi−1Γαi+βiΓαiΓβi−1∗1−ρini0ρini12πσXi2−ni1/2 exp−∑j:xij>0 lnXij−γi−12σXi2 ∑j:xij>0lnXij−γi−μXi2

(32) ∝σ ˆXi2−ni1/2 exp−∑j:xij>0 lnXij−γi−12σ ˆXi2 ∑j:xij>0lnXij−γi−μ ˆXi2

(33) fρi=Pρi|X=constant∗1−ρini0+αi−1ρini1βi−1Γαi+βiΓαiΓβi−1∗2πσXi2ni1/2 exp−∑j:xij>0 lnXij−γi−12σXi2 ∑j:xij>0lnXij−γi−μXi2

(34) ∝1−ρini0+αi−1ρini1+βi−1.

Next, we apply NA to the posterior distribution of κi = (μXi, logσXi); the logarithm of the posterior density is approximated by using a quadratic function of κi. The second derivatives of the log-posterior density are needed for constructing the approximation. From Eq. (3), the log-likelihood can be expressed as (35) lκi|X=ni0 ln1−ρi+ni1 lnρi−ni12ln2π+ni1 lnσXi− ∑j:xij>0 lnXij−γi−12σXi2ni1−1σ ˆXi2+ni1μ ˆXi−μXi2∝constant.+ni1 lnσXi−12exp2lnσXini1−1σ ˆXi2+ni1μ ˆXi−μXi2.

After that the first and second derivatives of (μXi, logσXi) respectively become

(36) ∂lκi|X∂μXi,∂2lκi|X∂μXi2=ni1μ ˆXi−μXiσXi2,−ni1σXi2

(37) ∂lκi|X∂lnσXi,∂2lκi|X∂lnσXi2=−ni1+σXi−2ni1−1σ ˆXi2+ni1μ ˆXi−μXi2,−2σXi−1ni1−1σ ˆXi2+ni1μ ˆXi−μXi2.

The point estimates of (μXi, logσXi) are derived after setting their first derivatives to zero. Meanwhile, the variances of their estimates are respectively obtained by using an inverse Fisher information matrix as follows:

(38) μ ˆXi lnσ ˆXi=μ ˆXi lnni1−1ni1−1σ ˆXi2

(39) Vμ ˆXiVlnσ ˆXi=σ ˆXi2/ni12ni1.

After using the Jacobian to back-transform from lnσ ˆXi to σXi2, the posterior densities of (μXi,σXi2) are respectively approximated as normal distribution as follows:

(40) fμXi=pμXi|X,σ ˆXi2,γ ˆi=12πσXi2/ni11/2 exp−ni12σXi2lnXij−γ ˆi−μ ˆXi2

(41) fσXi2=pσXi2|X=ni1+24πσ ~Xi41/2 exp−ni1+24σ ~Xi4lnXij−γ ˆi−σ ~Xi22

where σ ~Xi2=ni1σ ˆXi2/ni1+2. Therefore, the posterior density of ω becomes (42) fω=fρ1fγ1+ expfμX1fρ2fγ2+ expfμX2

Finally, the 100(1 − φ)% Bayesian-based CI for ω is (43) lωB,uωB=fωφ/2,fω1−φ/2

where fω(φ) denotes the φth percentile of fω. The CP of Bayesian CI (CPB) can be computed by using Algorithm 5.

The Monte Carlo simulation study

The comparative performances of the proposed Bayesian-, the fiducial GPQ- and MOVER-fiducial GPQ-, and NA-based CI were evaluated via a Monte Carlo simulation study. The settings for the simulation parameters for the two simulation studies were as follows. For each specified parameter combination, the 95% CIs for the ratio of the medians (ω) of TPLN distributions containing zero values were constructed based on 5000 randomly generated samples. In addition, 2500 Monte Carlo sampling passes were used for each generated sample for the fiducial GPQ-based method. To assess the performances of the methods, their CPs were calculated by using the proportion of 5000 simulated CIs covering ω. LEP and UEP are defined as the proportion of times that ω falls below and above the stimulated CIs, respectively. At the 95% nominal confidence level, the expected lengths (ELs) of the CIs is also needed for deciding which method performs the best. A good performance will produce CP = 95% and LEP = UEP = 2.5%. Likewise, the comparison between LEP and UEP can be expressed in terms of the relative bias, which is defined as (44) RB=UEP−LEPUEP+LEP.

Thus, a good balance between LEP and UEP will produce a relative bias close to zero. Last, the best-performing method will provide the shortest EL.

In the first simulation study, we chose a small proportion of zeros and variance ((d1, d2) = (10%, 10%),  (10%, 30%) and σX12=σX22=1.25, respectively), the results of which provide insight into the sampling behavior of the CIs (Table 1 and Figs. 1, 2 and 3). It can be seen that although all of the methods generated CPs above or close to the nominal confidence level in almost all of the scenarios, the LEPs, UEPs, and ELs produced by the Bayesian method demonstrated its superiority. Thus, the Bayesian method performed the best in situations with a small proportion of zeros and sample variance except for with a large sample size, with the MOVER-fiducial GPQ method performing the best in that case.

Table 1 Monte Carlo simulation results from the simulation study 1: σX12=σX22=1.25.

Scenarios	(n1, n2)	(ρ1, ρ2)%	(γ1, γ2)	B	F	M	N	Relative bias	
				LEP	CP	UEP	EL	LEP	CP	UEP	EL	LEP	CP	UEP	EL	LEP	CP	UEP	EL	B	F	M	N	
1	(25,25)	(10,10)	(1,1)	0.72	98.30	0.98	163.37	0.18	99.52	0.30	200.56	0.06	99.48	0.46	189.01	0.34	99.18	0.48	211.70	0.15	0.25	0.77	0.79	
2			(1,3)	0.66	98.44	0.90	156.22	0.28	99.48	0.24	198.36	0.00	99.58	0.42	186.41	0.28	99.26	0.46	202.52	0.15	−0.08	1.00	1.00	
3			(3,5)	0.74	98.44	0.82	142.20	0.24	99.68	0.08	195.00	0.00	99.80	0.20	182.46	0.36	99.28	0.36	182.52	0.05	−0.50	1.00	1.00	
4		(10,30)	(1,1)	0.96	98.30	0.74	180.80	0.36	99.42	0.22	222.67	0.18	99.58	0.24	208.90	0.48	99.14	0.38	240.36	0.13	−0.24	0.14	1.00	
5			(1,3)	0.90	97.98	1.12	171.38	0.54	99.22	0.24	218.83	0.04	99.46	0.50	204.42	0.44	99.00	0.56	225.81	0.11	−0.38	0.85	1.00	
6			(3,5)	0.90	98.02	1.08	157.74	0.28	99.54	0.18	215.19	0.08	99.52	0.40	200.41	0.44	99.08	0.48	206.96	0.09	−0.22	0.67	1.00	
7	(25,50)	(10,10)	(1,1)	0.38	96.92	2.70	136.08	0.12	98.84	1.04	167.07	0.12	99.00	0.88	158.35	0.14	98.98	0.88	177.91	0.75	0.79	0.76	0.56	
8			(1,3)	0.48	96.26	3.26	132.95	0.12	98.94	0.94	167.09	0.04	98.64	1.32	158.41	0.26	98.42	1.32	173.17	0.74	0.77	0.94	1.00	
9			(3,5)	0.28	96.96	2.76	121.35	0.02	99.36	0.62	165.16	0.02	98.96	1.02	156.11	0.14	98.86	1.00	156.03	0.82	0.94	0.96	1.00	
10		(10,30)	(1,1)	0.70	96.48	2.82	146.90	0.20	98.60	1.20	179.03	0.12	98.90	0.98	169.60	0.36	98.80	0.84	190.57	0.60	0.71	0.78	1.00	
11			(1,3)	0.44	96.86	2.70	142.87	0.18	98.86	0.96	178.49	0.06	98.70	1.24	168.96	0.36	98.70	0.94	183.63	0.72	0.68	0.91	1.00	
12			(3,5)	0.54	96.94	2.52	130.98	0.16	99.28	0.56	176.39	0.02	98.96	1.02	166.45	0.32	98.88	0.80	167.46	0.65	0.56	0.96	1.00	
13	(50,50)	(10,10)	(1,1)	2.20	95.28	2.52	103.10	0.74	98.34	0.92	125.77	0.38	98.60	1.02	121.00	0.56	98.66	0.78	137.55	0.07	0.11	0.46	0.47	
14			(1,3)	2.30	95.34	2.36	99.18	1.12	98.40	0.48	125.60	0.18	98.80	1.02	120.90	0.46	98.90	0.64	131.36	0.01	−0.40	0.70	1.00	
15			(3,5)	2.20	95.40	2.40	91.07	0.70	99.10	0.20	126.29	0.00	98.96	1.04	121.84	0.48	99.02	0.50	119.67	0.04	−0.56	1.00	1.00	
16		(10,30)	(1,1)	2.32	96.16	1.52	116.99	0.90	98.52	0.58	140.93	0.46	98.80	0.74	135.19	0.64	99.04	0.32	153.10	0.21	−0.22	0.23	1.00	
17			(1,3)	2.44	95.88	1.68	111.47	1.14	98.36	0.50	140.01	0.18	98.92	0.90	134.02	0.70	98.72	0.58	145.15	0.18	−0.39	0.67	1.00	
18			(3,5)	2.52	95.76	1.72	102.94	0.88	98.86	0.26	139.91	0.04	99.06	0.90	134.30	0.66	98.74	0.60	132.03	0.19	−0.54	0.91	1.00	
19	(50,100)	(10,10)	(1,1)	1.28	92.60	6.12	86.81	0.36	97.48	2.16	106.31	0.28	97.58	2.14	102.76	0.42	98.28	1.30	116.03	0.65	0.71	0.77	0.07	
20			(1,3)	1.34	92.66	6.00	84.88	0.64	98.04	1.32	107.17	0.12	97.80	2.08	104.00	0.34	98.38	1.28	112.56	0.63	0.35	0.89	1.00	
21			(3,5)	1.18	92.76	6.06	77.90	0.28	98.80	0.92	108.69	0.00	97.64	2.36	105.99	0.18	98.52	1.30	102.12	0.67	0.53	1.00	1.00	
22		(10,30)	(1,1)	1.56	94.06	4.38	93.63	0.50	97.66	1.84	113.43	0.34	97.88	1.78	109.72	0.38	98.58	1.04	123.78	0.47	0.57	0.68	1.00	
23			(1,3)	1.56	94.08	4.36	91.27	0.80	97.98	1.22	114.13	0.18	97.98	1.84	110.65	0.34	98.66	1.00	119.67	0.47	0.21	0.82	1.00	
24			(3,5)	1.38	94.16	4.46	84.33	0.44	98.78	0.78	115.57	0.02	98.16	1.82	112.46	0.42	98.52	1.06	109.19	0.53	0.28	0.98	1.00	
25	(100,100)	(10,10)	(1,1)	3.94	92.68	3.38	66.72	1.40	97.36	1.24	82.27	0.66	98.10	1.24	80.42	0.96	98.42	0.62	90.22	0.08	−0.06	0.31	−0.05	
26			(1,3)	4.22	92.16	3.62	64.20	2.08	97.30	0.62	83.51	0.12	98.12	1.76	82.05	0.74	98.56	0.70	86.14	0.08	−0.54	0.87	1.00	
27			(3,5)	4.36	91.82	3.82	59.41	0.82	98.80	0.38	87.00	0.00	98.32	1.68	86.29	0.64	98.54	0.82	78.57	0.07	−0.37	1.00	1.00	
28		(10,30)	(1,1)	4.32	92.50	3.18	75.39	1.56	97.30	1.14	91.48	0.68	97.88	1.44	89.20	0.86	98.32	0.82	99.62	0.15	−0.16	0.36	1.00	
29			(1,3)	4.64	92.48	2.88	72.36	2.00	97.58	0.42	92.34	0.14	98.58	1.28	90.33	0.58	98.70	0.72	94.97	0.23	−0.65	0.80	1.00	
30			(3,5)	4.78	92.48	2.74	67.33	1.46	98.32	0.22	95.34	0.04	98.74	1.22	94.07	0.76	98.56	0.68	87.22	0.27	−0.74	0.94	1.00	
31	(100,200)	(10,10)	(1,1)	1.66	89.92	8.42	56.58	0.54	96.54	2.92	70.38	0.30	96.88	2.82	69.05	0.36	98.42	1.22	76.93	0.67	0.69	0.81	−0.44	
32			(1,3)	2.38	90.08	7.54	55.34	1.14	97.62	1.24	72.25	0.18	97.10	2.72	71.61	0.64	98.02	1.34	74.88	0.52	0.04	0.88	1.00	
33			(3,5)	2.14	90.02	7.84	51.27	0.62	98.86	0.52	76.67	0.00	97.22	2.78	76.83	0.48	97.98	1.54	68.25	0.57	−0.09	1.00	1.00	
34		(10,30)	(1,1)	2.42	91.16	6.42	61.27	0.90	97.04	2.06	75.23	0.34	97.48	2.18	73.82	0.58	98.46	0.96	81.88	0.45	0.39	0.73	1.00	
35			(1,3)	2.64	90.74	6.62	59.68	1.28	97.66	1.06	76.81	0.18	97.06	2.76	75.98	0.50	98.16	1.34	79.33	0.43	−0.09	0.88	1.00	
36			(3,5)	3.00	90.52	6.48	55.51	0.88	98.46	0.66	80.91	0.04	97.10	2.86	80.90	0.76	97.88	1.36	72.88	0.37	−0.14	0.97	1.00	
Notes.

B the Bayesian CI

F the fiducial GPQ CI

M the MOVER-fiducial GPQ CI

N the NA CI

Figure 1 Simulation 1: Box plots of the coverage percentages for 36 scenarios based on the four methods: σX12=σX22=1.25 in the following cases [sample sizes, zero proportions]: (A)[(25,25),(10,10)%],(B)[(25,50),(10,10)%], (C)[(50,50),(10,10)%],(D)[(50,100),(10,10)%], (E)[(100,100),(10,10)%],(F)[(100,200),(10,10)%], (G)[(25,25),(10,30)%],(H)[(25,50),(10,30)%], (I)[(50,50),(10,30)%],(J)[(50,100),(10,30)%], (K)[(100,100),(10,30)%],(L)[(100,200),(10,30)%].

Figure 2 Simulation 1: Scatter plots of the relative bias for Scenarios 1–36 based on the four methods: σX12=σX22=1.25.

Figure 3 Simulation 1: Line plots of the expected width for 36 scenarios based on the four methods: σX12=σX22=1.25 in the following cases [sample sizes, zero proportions]: (A)[(25,25),(10,10)%],(B)[(25,50),(10,10)%], (C)[(50,50),(10,10)%],(D)[(50,100),(10,10)%], (E)[(100,100),(10,10)%],(F)[(100,200),(10,10)%], (G)[(25,25),(10,30)%],(H)[(25,50),(10,30)%], (I)[(50,50),(10,30)%],(J)[(50,100),(10,30)%], (K)[(100,100),(10,30)%],(L)[(100,200),(10,30)%].

In the next simulation, we were interested in scenarios with a large proportion of zeros and variance (d1 = d2 = (20%, 40%), (40%, 40%) and σX12=σX22=3, respectively) (Table 2 and Figs. 4, 5 and 6). Once again, the Bayesian method provided acceptable CPs, as well as better ELs and a better balance between LEP and UEP, than the other methods.

Application of the methods to compare hourly wind speed data from two areas in northern Thailand

Due to the rapid effects of climate change, agricultural growth, and the social economy, seasonal air pollution from the burning of agricultural waste in preparation for planting, forest fires, and waste disposal during the transition from the winter to the dry season are important factors that influence the environment in northern Thailand (IQAir, 2021). Wind can affect the movement of PM2.5 and PM10 when its speed is 7.2 km/hr or higher (Liu et al., 2020). When cold air mass moves from China to Thailand, the upper region of Thailand can potentially become very cold toward the end of winter, which reduces the northeast monsoon to a calm wind. For this reason, PM2.5 levels usually increase during the transition between the winter season to the dry season (Teerasuphaset & Culp, 2020). Phitsanulok is a city in lower northern Thailand about halfway between Chiang Mai and Bangkok where crop and forestland burning is extensive, resulting in extreme PM2.5 occurrences (IQAir, 2022),while Phayao is one of the three highest-ranking provinces for PM2.5 in the upper northern region (Group, 2021).

We used datasets of the hourly wind speed from Phitsanulok and Phayao (Table 3) recorded in January 2021 to illustrate the efficacies of our proposed methods for formulating CIs for the ratio of the medians of TPLN distributions containing zero values. The data were taken from the Thai Meteorological Department Automatic Weather System (Thai Meteorlogical Department Automatic Weather System, 2022). Since wind speed observations are always non-negative, they are suitable for fitting to the following distributions: Cauchy, chi-squared, exponential, lognormal, TPLN, logistic, normal, and t-distributions. The Akaike information criterion (AIC) can be used to determine the best-fitting distribution.

Table 2 Monte Carlo simulation results from the simulation study 2: σX12=σX22=3.

Scenarios	(n1, n2)	(ρ1, ρ2)%	(γ1, γ2)	B	F	M	N	Relative bias	
				LEP	CP	UEP	EL	LEP	CP	UEP	EL	LEP	CP	UEP	EL	LEP	CP	UEP	EL	B	F	M	N	
37	(25,25)	(20,40)	(1,1)	1.70	97.30	1.00	259.44	1.28	98.10	0.62	293.83	1.52	97.88	0.60	280.54	0.40	99.30	0.30	444.30	−0.26	−0.35	−0.43	0.89	
38			(1,3)	1.36	97.60	1.04	245.93	1.08	98.52	0.40	286.48	0.92	98.70	0.38	269.61	0.46	99.30	0.24	417.03	−0.13	−0.46	−0.42	1.00	
39			(3,5)	1.44	97.24	1.32	224.79	1.04	98.40	0.56	276.06	0.46	98.92	0.62	255.06	0.34	99.10	0.56	380.39	−0.04	−0.30	0.15	1.00	
40		(40,40)	(1,1)	1.66	97.02	1.32	283.20	1.00	98.28	0.72	323.39	1.38	97.84	0.78	307.55	0.24	99.36	0.40	516.55	−0.11	−0.16	−0.28	1.00	
41			(1,3)	0.78	98.06	1.16	270.82	0.42	99.00	0.58	317.18	0.50	98.72	0.78	297.65	0.20	99.52	0.28	486.72	0.20	0.16	0.22	1.00	
42			(3,5)	1.48	97.20	1.32	248.95	0.82	98.58	0.60	306.52	0.46	98.88	0.66	282.32	0.36	99.40	0.24	445.82	−0.06	−0.15	0.18	1.00	
43	(25,50)	(20,40)	(1,1)	1.22	96.54	2.24	213.10	0.92	97.48	1.60	235.89	1.06	97.38	1.56	227.43	0.44	99.02	0.54	320.58	0.29	0.27	0.19	0.73	
44			(1,3)	1.36	95.90	2.74	205.82	1.16	97.12	1.72	232.74	0.92	97.18	1.90	222.39	0.64	98.62	0.74	309.26	0.34	0.19	0.35	1.00	
45			(3,5)	1.14	96.56	2.30	188.47	0.82	97.82	1.36	225.40	0.42	98.10	1.48	211.28	0.26	99.02	0.72	283.15	0.34	0.25	0.56	1.00	
46		(40,40)	(1,1)	1.00	96.68	2.32	242.04	0.70	97.66	1.64	272.84	0.82	97.60	1.58	260.74	0.40	98.94	0.66	404.46	0.40	0.40	0.32	1.00	
47			(1,3)	0.70	96.42	2.88	235.54	0.56	97.68	1.76	270.25	0.48	97.60	1.92	256.41	0.34	98.80	0.86	395.13	0.61	0.52	0.60	1.00	
48			(3,5)	1.04	96.58	2.38	214.38	0.72	97.78	1.50	259.15	0.48	97.84	1.68	241.05	0.38	98.86	0.76	360.16	0.39	0.35	0.56	1.00	
49	(50,50)	(20,40)	(1,1)	1.96	96.32	1.72	177.64	1.52	97.26	1.22	193.23	1.60	97.12	1.28	187.25	0.90	98.70	0.40	249.14	−0.07	−0.11	−0.11	0.67	
50			(1,3)	2.02	96.36	1.62	168.55	2.06	96.98	0.96	189.16	1.28	97.54	1.18	180.85	0.76	98.72	0.52	237.29	−0.11	−0.36	−0.04	1.00	
51			(3,5)	1.88	95.98	2.14	155.15	1.64	97.46	0.90	184.51	0.74	98.02	1.24	173.93	0.76	98.64	0.60	218.59	0.06	−0.29	0.25	1.00	
52		(40,40)	(1,1)	1.50	97.02	1.48	192.68	1.20	97.58	1.22	209.88	1.20	97.64	1.16	203.27	0.54	98.90	0.56	273.71	−0.01	0.01	−0.02	1.00	
53			(1,3)	1.96	96.44	1.60	184.84	2.00	97.12	0.88	206.78	1.26	97.58	1.16	198.06	0.96	98.42	0.62	261.51	−0.10	−0.39	−0.04	1.00	
54			(3,5)	1.48	96.80	1.72	169.90	1.28	97.74	0.98	200.86	0.58	98.20	1.22	189.24	0.52	98.74	0.74	241.20	0.08	−0.13	0.36	1.00	
55	(50,100)	(20,40)	(1,1)	1.56	96.02	2.42	147.79	1.12	97.06	1.82	159.74	1.24	97.00	1.76	155.22	0.52	98.76	0.72	202.75	0.22	0.24	0.17	0.34	
56			(1,3)	1.30	96.16	2.54	142.96	1.26	97.20	1.54	158.18	0.74	97.24	2.02	152.63	0.48	98.98	0.54	196.49	0.32	0.10	0.46	1.00	
57			(3,5)	1.36	96.22	2.42	131.20	1.24	97.54	1.22	154.66	0.46	97.88	1.66	147.32	0.42	98.82	0.76	180.39	0.28	−0.01	0.57	1.00	
58		(40,40)	(1,1)	1.38	96.32	2.30	165.56	1.06	96.80	2.14	179.74	1.10	96.86	2.04	174.31	0.30	98.60	1.10	230.58	0.25	0.34	0.30	1.00	
59			(1,3)	1.16	96.36	2.48	160.90	1.16	97.36	1.48	178.08	0.70	97.36	1.94	171.83	0.48	98.48	1.04	223.81	0.36	0.12	0.47	1.00	
60			(3,5)	1.32	96.10	2.58	148.06	1.14	97.42	1.44	173.56	0.50	97.54	1.96	164.89	0.60	98.46	0.94	206.05	0.32	0.12	0.59	1.00	
61	(100,100)	(20,40)	(1,1)	2.40	95.50	2.10	122.51	1.98	96.50	1.52	131.58	1.68	96.78	1.54	128.30	0.70	98.56	0.74	165.11	−0.07	−0.13	−0.04	0.25	
62			(1,3)	2.28	95.80	1.92	116.67	2.30	96.74	0.96	129.62	1.14	97.30	1.56	125.17	0.86	98.54	0.60	156.54	−0.09	−0.41	0.16	1.00	
63			(3,5)	2.30	95.52	2.18	107.65	2.20	96.92	0.88	128.12	0.48	98.06	1.46	122.77	0.86	98.46	0.68	143.65	−0.03	−0.43	0.51	1.00	
64		(40,40)	(1,1)	2.20	95.52	2.28	133.62	1.78	96.48	1.74	143.42	1.64	96.46	1.90	139.78	0.72	98.46	0.82	180.55	0.02	−0.01	0.07	1.00	
65			(1,3)	1.88	95.86	2.26	128.21	2.00	96.68	1.32	141.60	0.94	97.30	1.76	136.82	0.68	98.54	0.78	172.85	0.09	−0.20	0.30	1.00	
66			(3,5)	1.58	95.90	2.52	118.25	1.56	97.06	1.38	139.27	0.40	97.62	1.98	133.33	0.44	98.66	0.90	158.17	0.23	−0.06	0.66	1.00	
67	(100,200)	(20,40)	(1,1)	1.58	95.16	3.26	101.46	1.34	96.36	2.30	108.80	1.24	96.34	2.42	106.37	0.64	98.36	1.00	134.21	0.35	0.26	0.32	−0.20	
68			(1,3)	1.46	95.54	3.00	98.30	1.70	96.74	1.56	108.44	0.88	96.68	2.44	105.68	0.40	98.50	1.10	130.20	0.35	−0.04	0.47	1.00	
69			(3,5)	1.68	95.32	3.00	90.42	1.54	97.32	1.14	107.71	0.36	97.30	2.34	104.56	0.56	98.46	0.98	119.13	0.28	−0.15	0.73	1.00	
70		(40,40)	(1,1)	1.40	95.62	2.98	114.31	1.04	96.50	2.46	122.60	1.02	96.54	2.44	119.66	0.38	98.56	1.06	152.21	0.36	0.41	0.41	1.00	
71			(1,3)	1.28	95.74	2.98	111.45	1.54	96.78	1.68	122.19	0.68	96.70	2.62	118.93	0.36	98.48	1.16	148.24	0.40	0.04	0.59	1.00	
72			(3,5)	1.58	95.00	3.42	102.93	1.24	97.16	1.60	121.02	0.48	96.70	2.82	116.90	0.52	98.26	1.22	135.95	0.37	0.13	0.71	1.00	
Notes.

B the Bayesian CI

F the fiducial GPQ CI

M the MOVER-fiducial GPQ CI

N the NA CI

Figure 4 Simulation 2: Box plots of the coverage percentages for Scenarios 37-72 based on the four methods: σX12=σX22=3 in the following cases [sample sizes, zero proportions]:(A)[(25,25),(20,40)%], (B)[(25,50),(20,40)%], (C)[(50,50),(20,40)%],(D)[(50,100),(20,40)%], (E)[(100,100),(20,40)%],(F)[(100,200),(20,40)%], (G)[(25,25),(40,40)%],(H)[(25,50),(40,40)%], (I)[(50,50),(40,40)%],(J)[(50,100),(40,40)%], (K)[(100,100),(40,40)%],(L)[(100,200),(40,40)%].

Figure 5 Simulation 2: Scatter plots of the relative bias for Scenarios 37-72 based on the four methods: σX12=σX22=3.

Figure 6 Simulation 2: Line plots of the expected width for 36 scenarios based on the four methods: σX12=σX22=3 in the following cases [sample sizes, zero proportions]: (A)[(25,25),(20,40)%],(B)[(25,50),(20,40)%], (C)[(50,50),(20,40)%],(D)[(50,100),(20,40)%], (E)[(100,100),(20,40)%],(F)[(100,200),(20,40)%], (G)[(25,25),(40,40)%],(H)[(25,50),(40,40)%], (I)[(50,50),(40,40)%],(J)[(50,100),(40,40)%], (K)[(100,100),(40,40)%],(L)[(100,200),(40,40)%].

Table 3 Data of hourly wind speed (km/3 h) in Phisanulok and Phayao provinces, northern Thailand during January 1–15, 2021.

Phisanulok	Phayao	
19.9	0	4.3	13.7	9.4	0	26.7	15	
20.6	0	21.6	21.9	16.1	2.2	20.9	13.3	
26	11.6	12.6	19.1	17.3	7.5	19.1	9.7	
9.4	22.7	18.4	21.2	8.6	15.8	20.5	14	
9.8	16.5	16.6	20.1	2.9	14.5	21.7	11.9	
12.6	6.1	23.3	20.2	1.8	4.3	29.9	11.1	
11.6	0	31	9	0	2.9	33.1	14.3	
0	6.8	6.1	22	1.4	4.3	23	12.3	
20.9	3.6	1.8	20.9	9.7	2.2	22.6	17.2	
14.8	1.5	5	12.6	23.4	0	29.9	6.1	
27.3	9	1.8	0	19.8	5.4	24.1	4.7	
13	10.4	0	0	3.9	12.6	16.2	3.7	
0	18.3	14.4	4.3	1.1	15.4	32	6.1	
5.4	8.3	14.5	6.9	0	6.2	29.4	4.6	
3.2	0	24.4	16.6	0	3.2	14	11.1	
8.7	0.4	6.9	15.9	1.8	2.9	25.2	23.4	
8.7	0	2.5	11.6	4.3	0.4	30.2	21.2	
13	0	4.3	6.9	22.3	0.4	23.4	4.7	
25.9	8.7	2.2	0	21.2	10.8	31.6	1.8	
8	25.9	2.5	0	7.6	12.2	29.5	4.7	
5	21.7	15.8	0	0.4	21.6	20.1	1.4	
3.6	8.3	15.2	2.1	1.1	14.8	30.6	0	
5.7	4.7	14.4	4.3	0	13	18.4	7.2	
6.2	4.7	7.2	14.4	0.7	17.6	21.2	17.3	
7.6	2.2	15.8	19.5	7.2	16.9	14.1	16.6	
13	10.1	26.3	5	15.5	2.1	14.8	3.9	
25.6	14	32	0	16.2	8.2	19.1	3.2	
15.1	20.5	32.7	4.7	5	21.9	25.3	1.8	
2.2	17.6	32.4	0	0	21.3	24.5	0.7	
3.6	5.8	20.2	4	0	23.5	19.5	0	
Notes.

Source: Thai Meteorological Department Automatic Weather System.

URL: http://www.aws-observation.tmd.go.th/web/reports/weather_minute.asp.

We found that the TPLN model was suitable for the wind speed data, as evidenced by its smallest AIC value in Table 4. The basic statistics for the datasets are reported in Table 5. By way of comparison, the ratio of the medians of the TPLN distributions of the hourly wind speed data from Phayao and Phitsanulok is ω ˆ=1.0217 for medians η ˆphayao=7.4091 and η ˆphitsanulok=7.2514. The 95% CIs and corresponding lengths based on the fiducial, NA, and Bayesian methods for the ratio of the medians of TPLN distributions containing zero values are reported in Table 6.

Table 4 AIC results for the positive wind speed data (km/3 h) in Phisanulok and Phayao provinces, northern Thailand during January 1–15, 2021.

Provinces	Cauchy	Chi-suqare	Exponential	Lognormal	TPLN	Logistic	Normal	T	
Phitsanulok	787.8140	818.3827	739.9249	733.9261	725.9633	742.5044	736.4505	736.7834	
Phayao	865.3148	1008.1470	793.3140	816.9264	800.6639	812.1632	803.3119	803.8294	
Notes.

Bold font denotes the best-fitting model for the data.

Table 5 Basic Statistics for the wind speed data.

Basic statistics	Provinces	
	Phitsanulok	Phayao	
Groups (i)	1	2	
sample sizes (ni)	120	120	
Mean (μ ˆi)	2.1581	2.0196	
Variances (σ ˆXi2)	1.4911	2.3389	
Zero proportions (ρ ˆi:%)	13.33	8.33	
Threshold (γ ˆi)	0.399	0.389	
Median wind speed (η ˆi)	8.0248	6.8204	
The ratio of the medians (ω ˆ=η ˆ1/η ˆ2)	ω ˆ=1.1765	

Table 6 The 95% CIs for the ratio of the median wind speed in Phisanulok and Phayao provinces.

Methods	95% CIs	Lengths	
Fiducial GPQ	(0.7825, 1.7937)	1.0112	
MOVER-fiducial GPQ	(0.7823, 1.7203)	0.9379	
NA	(0.6922, 1.9997)	1.3074	
Bayesian	(0.8047, 1.8238)	1.0190	

It can be interpreted that there is no difference between the wind speeds in Phayao and Phitsanulok. The majority of the population in both areas are agriculturists, and so agricultural burning is often carried out in preparation for planting and after harvesting. Furthermore, the empirical example results are in agreement with the Monte Carlo simulation results in the previous section; the EL of the MOVER-fiducial GPQ CI was the smallest with a suitable CP for a small variance and a large sample size. Overall, the Bayesian-based method is the most suitable for formulating CIs for the ratio of the medians of TPLN distributions containing zero values when taking the checking criteria results from scenarios 1–12 and 37–72 into account.

Discussion

We applied fiducial, NA, and Bayesian-based method to formulate CIs for the ratio of the medians of TPLN distributions containing zero values. From the results of the simulation study, the fiducial methods based on fiducial GPQ and MOVER-fiducial GPQ always provided CPs greater than the nominal 95% confidence level because the fiducial GPQ of (γi, ρi, μXi) have strong points (Hannig, Iyer & Patterson, 2006), as revealed by the conditions for FGPQ2 in Definition 1. However, the MOVER-fiducial GPQ method produced shorter interval than the fiducial GPQ and worked well for a small variance and a large sample size. The NA method provided CPs greater than the nominal 95% confidence level, thereby making its ELs longer than the other methods, which could have been caused by the variance in the estimated median (McKean & Schrader, 1984).

Meanwhile, the Bayesian method provided suitable CPs with the shortest interval length, except for a small variance and a large sample size for which the MOVER-fiducial GPQ method performed the best. This could be because of using the uniform prior to gain information about the parameters from the data to obtain their posterior density. As such, the constructed CIs for the ratio of the medians of TPLN distributions containing zero values based on the Bayesian method with a uniform prior performed well.

Conclusions

CIs for the ratio of the medians of TPLN distributions containing zero values were formulated by using fiducial-, NA-, and Bayesian-based methods. Since a theoretical comparison was not possible, a Monte Carlo simulation and empirical application with two real datasets of wind speed observations were used to evaluate their performances in terms of their CPs and ELs. The results of the simulation study led us to recommend the Bayesian method for constructing the CIs for the ratio of the medians of TPLN distributions containing zero values because it attained CPs close to the nominal 95% confidence level and the shortest EL in most cases, except for a small variance and a large sample size for which the MOVER-fiducial GPQ method should be used.

Supplemental Information

Supplemental Information 1 Data of hourly wind speed (km/3hr) in Phisanulok and Phayao provinces, northern Thailand between January 1-15, 2021

Click here for additional data file.

Supplemental Information 2 R code for the program: computes coverage probability and expected lengths for all results

Click here for additional data file.

Additional Information and Declarations

Competing Interests

Author Contributions

Data Availability

The authors declare there are no competing interests.

Patcharee Maneerat conceived and designed the experiments, performed the experiments, analyzed the data, prepared figures and/or tables, authored or reviewed drafts of the article, and approved the final draft.

Pisit Nakjai conceived and designed the experiments, performed the experiments, analyzed the data, prepared figures and/or tables, and approved the final draft.

Sa-Aat Niwitpong conceived and designed the experiments, authored or reviewed drafts of the article, and approved the final draft.

The following information was supplied regarding data availability:

The raw data is available in the Supplemental Files.

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
