# Peer review of "Estimation methods for the ratio of medians of three-parameter lognormal distributions containing zero values and their application to wind speed data from northern Thailand"

_PeerJ, doi:10.7717/peerj.14194_

## Round 0.1 · original submission · Major Revisions

Please take the help of professional editing services or consult some English-speaking colleagues to improve the write-up of the manuscript. Without a sound write-up, it is not publishable in its current form. Moreover, please quote some of the latest references to show the relevance of your study to this journal, and to cover the latest development in the topic.

Reviewer 1 ·

Basic reporting

The topics is important, the results are well thought-of and convincing, the paper should be, in my opinion, eventually published.

Experimental design

The authors apply known algorithms -- as well as a new algorithm -- to analyze experimental data.

Validity of the findings

The results are convincing and well justified.

Additional comments

From the viewpoint of mathematics and applications, the paper is very good. The weak point is English, many phrases need correction. The authors should ask someone who knows English better to help edit their text. In the present form, the paper cannot be published.

I will mention a few places where English needs to be improved, but there are many more.

line 40: transition ... is currently underway -- this phrase may refer to the moment when the authors submitted this paper, but in a published version, it makes no sense, since to the readers currently means at the moment when they read this paper.

line 80: the authors like the word Furthermore but do not understand its meaning, delete it. Also, while one of the methods is indeed proposed by the authors, all others are not, they were proposed earlier, I suggest simply These methods ...

line 95, 107, etc.: if the text continues after the formula, with words like where which etc., the corresponding line should not be indented

lines 99 and 100 incomprehensible; I suggest replacing "such that ... becomes" with "so", and deleting "which" on p. 100.

line 101: delete Importantly, and replace the unclear "it" with "Adam"

line 109: explores until they biased -- no idea what this means

formula (10): use \left( and \right) in LaTeX, to make sure that the parentheses include everything in between

line 114: it leads to gain -> one can compute

line 127: revealed -> described

lines 128-129: can be led to the development... -- I have no idea what is means, my guess is what they want to say is that fudicial techniques have been used in several applications

line 130: random variable of the probability function ??? probably random variable with probability density function

definitions: emphasize defined terms by placing them in a different font

line 138: is followed the condition -> satisfies the condition

it can be constructed the -> it is possible to construct

I can continue with many comments on every page -- and I did not even list all the comments on the first four pages. The authors need to take care of it, otherwise the paper cannot be published.

Reviewer 2 ·

Basic reporting

The text are written well and clear.

Experimental design

No comment

Validity of the findings

The results are practical.

Additional comments

1, Please explain why CI for the ratio of medians of three parameter lognormal distribution with excessive zeros could be the used model the wind speed.
2. Please explain how much it differs fundamentally from the authors' "Confidence intervals for rainfall dispersions using the ratio of two coefficients of variation of lognormal distributions with excess zeros".
3. The similarity between the rainfall data and wind speed data can be recognized. Please explain why the same model could be used for both rainfall and the windspeed data?

---

## Round 0.2 · Minor Revisions

Reviewer 2 has accepted the paper and Reviewer 1 suggest some minor comments.

Reviewer 1 ·

Basic reporting

The authors made practically all the changes recommended by the reviewer. Math was good already, so I think the paper can be accepted.

Minor thing: I recommended, in each definition, to place the defined term in a different font, to make it clear what exactly is defined. The authors clearly did not understand this suggestion, I would still like it to be implemented. Here is a LaTeX example of what I have in mind.

{\bf Definition.} {\em A number $x$ is called {\em positive} if $x>0$.}

This way, almost all the text of the definition is in italics, with the exception of the defined term "positive" which is in the regular non-italics font.

Experimental design

The authors use already available data.

Validity of the findings

Valid

Additional comments

None

Reviewer 2 ·

Basic reporting

This article is well written.

Experimental design

After the preliminary review and the comments submitted, this article has defined and addressed the problem professionally.

Validity of the findings

All the conclusion is well supported by the underlying theory. The results are convincing enough to be published in the journal.

---

## Round 0.3 · accepted · Accept

The authors have addressed all concerns of reviewers.